# Exploring the Role of Non-Coding RNAs in the Pathophysiology of Systemic Lupus Erythematosus

**DOI:** 10.3390/biom10060937

**Published:** 2020-06-22

**Authors:** Mohammad Taheri, Reyhane Eghtedarian, Marcel E. Dinger, Soudeh Ghafouri-Fard

**Affiliations:** 1Urogenital Stem Cell Research Center, Shahid Beheshti University of Medical Sciences, Tehran 1985717443, Iran; mohammad_823@yahoo.com; 2Department of Medical Genetics, Shahid Beheshti University of Medical Sciences, Tehran 1985717443, Iran; reghtedarian@gmail.com; 3School of Biotechnology and Biomolecular Sciences, University of New South Wales, Sydney, NSW 2052, Australia

**Keywords:** lncRNA, miRNA, systemic lupus erythematosus

## Abstract

Systemic lupus erythematosus (SLE) is a chronic immune-related disorder designated by a lack of tolerance to self-antigens and the over-secretion of autoantibodies against several cellular compartments. Although the exact pathophysiology of SLE has not been clarified yet, this disorder has a strong genetic component based on the results of familial aggregation and twin studies. Variation in the expression of non-coding RNAs has been shown to influence both susceptibility to SLE and the clinical course of this disorder. Several long non-coding RNAs (lncRNAs) such as GAS5, MALAT1 and NEAT1 are dysregulated in SLE patients. Moreover, genetic variants within lncRNAs such as SLEAR and linc00513 have been associated with risk of this disorder. The dysregulation of a number of lncRNAs in the peripheral blood of SLE patients has potentiated them as biomarkers for diagnosis, disease activity and therapeutic response. MicroRNAs (miRNAs) have also been shown to affect apoptosis and the function of immune cells. Taken together, there is a compelling rationale for the better understanding of the involvement of these two classes of non-coding RNAs in the pathogenesis of SLE. Clarification of the function of these transcripts has the potential to elucidate the molecular pathophysiology of SLE and provide new opportunities for the development of targeted therapies for this disorder.

## 1. Introduction

Systemic lupus erythematosus (SLE) is a chronic immune-related disorder described by a lack of tolerance to self-antigens and the over-secretion of autoantibodies against host DNA and other cellular elements. This disorder mostly affects females of reproductive age. SLE has a broad range of clinical manifestations that affect several organs. Approximately 50% of SLE cases will have serious and life-threatening complications such as nephritis, vasculitis, pulmonary hypertension, interstitial lung disorder and cerebral stroke [1]. The significantly higher concordance rate in monozygotic twins (24–35%) compared with what has been reported in dizygotic twins (2–5%) [2,3] indicates the presence of a genetically determined component in this disorder. Several genetic loci such as the major histocompatibility complex [4], Fc gamma (Fcγ) receptors [5] and the signal transducer and activator of transcription (STAT) genes [6] have been associated with risk of SLE in different populations. However, it is estimated that the majority of the genetic variation acknowledged until now illuminates less than 10% of genetic SLE susceptibility [7]. Recent studies have shown the prominent roles of non-coding RNAs in the pathogenesis of SLE [1]. These transcripts comprise a large proportion of the whole transcriptome and exert regulatory roles on specific targets or large regions of the genome [8]. Classified into two main groups comprising either long non-coding RNAs (lncRNAs) or microRNAs (miRNAs) based on their sizes, non-coding RNAs are involved in the regulation of critical biological processes including cell differentiation [8]. The GENCODE v7 catalog of human lncRNAs has reported 9277 annotated genes, generating a total of 14,880 transcripts [9]. Subsequent studies have led to the annotation of more than 172,000 lncRNA transcripts in the human genome [10]. Moreover, a recent combination of in silico high- and experimental low-throughput methods has estimated the presence of 2300 mature miRNAs in humans [11]. In the current manuscript, we describe the role of lncRNAs and miRNAs in the pathogenesis of SLE.

## 2. LncRNAs in SLE

LncRNAs are typically defined as transcripts longer than 200 nucleotides in length with no recognizable protein-coding potential. Approximately three decades have passed since the discovery of the first lncRNA [12]. Based on the information provided by the NONCODE database, there are more than 172,000 annotated lncRNA transcripts in the human genome [10]. These transcripts have known functions in the immune cell differentiation and the regulation of immune responses [13]. The regulatory roles of lncRNAs can be exerted through both cis- and trans-mechanisms. The former are best exemplified in enhancer-associated RNAs [14]. The interaction between lncRNAs and transcription factors and the subsequent construction of the preinitiation complex (PIC) can activate or suppress gene transcription [15]. Being formed by almost 100 proteins, the PIC has a crucial role in the transcription of protein-coding genes. The PIC recruits RNA polymerase II at gene transcription start sites and positions DNA into the RNA polymerase II active site for transcription [16]. Moreover, some lncRNAs can recruit co-activators to modulate the chromatin looping of adjacent genes, enhancing the transcription of these genes [17].

Several studies have noted the dysregulation of lncRNAs in SLE patients. One study comparing transcriptomes between SLE patients and health controls identified the differential expression of 163 lncRNAs [18]. Real-time PCR verified the upregulation of ENST00000604411.1 and ENST00000501122.2 and the downregulation of lnc-HSFY2-3:3 and lnc-SERPINB9-1:2 in the monocyte-derived dendritic cells (moDCs) of SLE patients. Notably, the expression levels of ENST00000604411.1 and ENST00000501122.2 were associated with disease activity [18]. An integrative assessment of the expression profiles of lncRNAs and messenger RNAS (mRNAs) in the peripheral blood mononuclear cells (PBMCs) of SLE patients led to the identification of the differential expression of more than 8,000 lncRNAs and more than 6,000 mRNAs between SLE patients and normal controls. Among the hundreds of concordantly-expressed lncRNA–mRNA pairs, most were enriched in transcriptional dysregulation in cancer and amino acid degradation pathways [19]. In a study identifying the biomarkers for SLE, the assessment of the levels of lncRNAs in plasma revealed lower levels of GAS5 and lnc-DC and elevated levels of linc0597 in SLE patients [20]. Notably, the expression of lnc-DC was higher in SLE patients with lupus nephritis (LN) compared with those without nephritis. Assessment of the diagnostic power of lncRNAs led to the recognition of GAS5, linc0597 and lnc-DC as specific markers for SLE. The combination of GAS5 and linc0597 was found to enhance diagnostic power. Moreover, lnc-DC has been suggested as a marker for the identification of nephritis in SLE patients [20]. In an in silico study examining the expression profile of lncRNAs and mRNAs from the Gene Expression Omnibus dataset, numerous interaction pairs of lncRNA-adjacent targeted mRNAs were identified, including NRIR-RSAD2, RP11-153M7.5-TLR2, RP4-758J18.2-CCNL2, RP11-69E11.4-PABPC4 and RP11-496I9.1-IRF7/HRAS/PHRF1 [21]. Notably, mitogen-activated protein kinase (MAPK) was found among the enriched signaling pathways of differentially expressed transcripts [21].

The well-characterized lncRNA NEAT1 has been found to be upregulated in SLE patients [22]. NEAT1 was mainly expressed in human monocytes and its expression was stimulated by LPS through a p38-mediated pathway. The downregulation of NEAT1 leads to suppression of the expression of several chemokines and cytokines such as IL-6 and CXCL10. NEAT1 participates in TLR4-associated inflammatory responses by influencing the activity of the MAPK-related axis. Notably, the expression of NEAT1 was found to be correlated with the course of SLE [22].

A number of other previously identified lncRNAs have also been found to be dysregulated in SLE patients. For example, lnc5150 was found to be downregulated in the PBMCs of SLE patients compared with healthy individuals. Moreover, lnc3643 expression was downregulated in SLE patients with proteinuria compared with patients without this symptom. The expression of lnc7514 was decreased in patients who tested positive for anti-double-stranded DNA (anti-dsDNA) antibodies compared with those that tested negative. The expression of lnc3643 was correlated with CRP and ESR levels. Finally, the expression of lnc7514 was associated with SLE activity and ESR levels [23]. Comparing lncRNAs in the PBMCs of SME patients with rheumatoid arthritis (RA) and healthy individuals, it was found that linc0949 and linc0597 were downregulated [24]. Notably, the expression of linc0949 was correlated with disease activity and the concentrations of complement component C3. Moreover, the expression of linc0949 was decreased in SLE patients with a history of organ damage and a significant association was found between the downregulation of linc0949 and the presence of LN. In the same study, appropriate treatment of SLE patients was found to enhance linc0949 expression. These results indicate the possibility that linc0949 could serve as a diagnostic marker for SLE that correlates with disease activity [24]. In an animal model of SLE, the suppression of the NF-κB pathway was shown to exert a protective role against kidney injury [25]. This effect was suggested to be a result of enhancement of the expression of the lncRNA TUG1 [25]. TUG1 was also found to be downregulated in the PBMCs of SLE patients compared with controls. The expression of this lncRNA was particularly downregulated in SLE patients with lupus nephritis. The authors detected negative correlations between the expression of this lncRNA and the SLE Disease Activity Index score, ESR, disease duration and proteinuria. Moreover, the expression of TUG1 was associated with complement C3 levels. The expression levels of TUG1 could effectively differentiate SLE patients with lupus nephritis [26]. A microarray-based study of lncRNA signatures in the T cells of SLE patients and normal persons led to the identification of more than 1900 dysregulated lncRNAs; among them were uc001ykl.1 and ENST00000448942, which were down-regulated in SLE patients. The expression of uc001ykl.1 was associated with ESR and CRP, while the expression of ENST00000448942 was associated with ESR and anti-Smith antibodies. Moreover, the expression of a number of lncRNAs was correlated with the activity of SLE in the affected individuals. LncRNA–mRNA co-expression assessments revealed a putative competing endogenous RNA role for these lncRNAs [27].

Figure 1 shows the molecular mechanism of the involvement of NEAT1 in the pathogenesis of SLE.

Table 1 and Table 2 show the list of down- and upregulated lncRNAs in SLE patients, respectively.

Notably, a number of studies have assessed the association between single nucleotide polymorphisms (SNPs) within lncRNAs and SLE. A recent genome-wide analysis of SLE-associated polymorphisms in lncRNA gene loci has shown an association between the rs13259960 in the SLEAR lncRNA and susceptibility to SLE. This variant is located in an intronic enhancer and changes the recruitment of STAT1 to the enhancer. Thus, the minor allele of this SNP diminishes SLEAR expression. SLEAR interacts with ILF2, hnRNP F and TAF15 to construct a complex that enhances the transcription of the anti-apoptotic genes. The role of this SNP in the regulation of apoptosis has also been verified in clinical samples [28]. The assessment of transcript signatures in SLA patients identified the lcnRNA linc00513 as significantly upregulated relative to healthy controls [32]. Interestingly, linc00513, which has also been recognized as a regulator of the type I interferon pathway [32], encompasses a functional risk locus for SLE in its promoter region. Notably, the risk alleles of the rs205764 and rs547311 SNPs increase the promoter activity of linc00513, resulting in the over-expression of this lncRNA in SLA patients. Table 3 shows the summary of studies which assessed the association between lncRNA SNPs and SLE.

## 3. MiRNAs in SLE

In an animal model of SLE, similar dysregulation of a number of miRNAs was observed in both PBMCs and splenocytes [33], leading to the suggestion that the miRNA profile of PBMCs reflects their signature in the lymphoid organ spleen. This finding potentiates PBMCs as a biological source for observing the kinetics of SLE-related miRNAs during the course of the disorder or after starting therapeutic modalities [33]. Assessment of the expression of B cell-related miRNAs in plasma samples from SLE patients revealed the downregulation of 14 miRNAs in SLE patients compared with healthy subjects [34]. Moreover, the expression of six miRNAs was substantially lower in SLE patients compared with RA patients. MiRNA signatures could not only differentiate SLE patients from healthy subjects, but could also discriminate SLE patients from RA patients. Furthermore, a particular miRNA profile including miR-19b, miR-25, miR-93 and miR-15b could predict disease activity in SLE patients. Independently, the expression levels of miR-15b have also been previously associated with LN [34]. In a study examining a panel of differentially expressed miRNAs in SLE patients with and without renal involvement, hsa-miR-766-3p, which participates in the regulation of the PI3K-AKT-mTOR pathway, was identified as significantly dysregulated [35]. The authors reported the significance of proinflammatory cytokines, disease stage and severity in the observed differential expression of miRNAs. In the same study, hsa-miR-621 was also suggested to contribute to the pathogenesis of hypertension in SLE [35]. The dysregulation of miR-145, miR-224, miR-513-5p, miR-150, miR-516a-5p, miR-483-5p and miR-629 has been indicated in the T cells of SLE patients, with downregulation of miR-145 and the upregulation of miR-224 subsequently validated in SLE patients [36]. These miRNAs have been shown to target STAT1 and apoptosis inhibitory protein 5 (API5), respectively. In vitro studies indicated that abnormal expression of these miRNAs facilitates T cell activation-induced cell death, and over-expression of STAT1 was linked with lupus nephritis in SLE patients [36]. A separate study reported the downregulation of miR-410 in the T cells of SLE patients [37]. The upregulation of miR-410 has been shown to decrease the expression of IL-10. This miRNA inhibits the expression of STAT3 through interaction with its 3’ untranslated region (UTR). Knockdown of STAT3 decreased the expression of IL-10 in CD3+ T cells. Thus, miR-410 is involved in the pathophysiology of SLE through modulating the expression of IL-10 via the STAT3 axis [37]. The miRNA miR-29b has been found to be upregulated in SLE patients [38]. Notably, its expression was negatively correlated with the expression of Sp1 and DNMT1. The upregulation of miR-29b in CD4+ T cells enhanced hypomethylation and the expression of the CD11a and CD70 genes, while the suppression of miR-29b had the opposite effect. Therefore, miR-29b decreases DNMT1 expression through the modulation of Sp1 in T cells [38]. In bone marrow mesenchymal stem cells of SLE patients, a negative association between the expression of let-7f and disease activity was identified [39]. The downregulation of let-7f decreased the proliferation of these cells, leading to the upregulation of Th17 cells and the downregulation of Treg cells through targeting IL-6 and inducing the STAT3 axis [39]. Yuan et al. indicated decreased levels of miR-98 in the PBMCs of SLE patients, and negative association between its level and IL-6 concentration. The expression of this miRNA in SLE patients was associated with SLE activity, lupus nephritis, and the presence of the anti-dsDNA antibody. This miRNA is predicted to influence the expression of IL-6. IL-6 upregulation enhances the proliferation of PBMCs and facilitates the production of pro-inflammatory cytokines. MiR-98 modulates STAT3 phosphorylation through the regulation of IL-6 expression. Thus, miR-98 can amend STAT3-associated cell proliferation and the release of pro-inflammatory cytokines in patients with SLE [40]. The downregulation of miR-98 has also been reported in SLE CD4+ T cells, while Fas levels were enhanced [41]. In vitro studies have shown that the over-expression of miR-98 secured Jurkat cells against Fas-associated apoptosis. Direct interaction between miR-98 and Fas mRNA has been verified through functional studies. Thus, the under-expression of miR-98 enhances apoptosis by altering the Fas-mediated apoptotic axis in SLE CD4+ T cells [41]. Downregulation of miR-125a and the upregulation of its target gene KLF13 has been reported in SLE patients [42]. The upregulation of miR-125a decreases the expression of RANTES and KLF13. This miRNA downregulates the expression of RANTES through targeting KLF13 [42]. A microarray-based analysis has shown the dysregulation of 37 miRNAs in SLE patients, among them the down-regulated miR-125b whose expression was negatively correlated with lupus nephritis. This miRNA has been shown to target ETS1 and STAT3. Their high throughput analysis has shown that the signature of this miRNA can be regarded as a potential biomarker of SLE. Besides, the deceased levels of miR-125b in T cells participate in the development of SLE through modulating the ETS1 and STAT3 levels [43]. Downregulation of the estrogen-regulated miRNA miR-302d has been observed in SLE patient monocytes [44]. This miRNA regulates the expression of interferon regulatory factor (IRF) 9. Mechanistically, miR-302d has a protective effect against pristane-associated inflammation in animal models through influencing the expression of IRF9 and ISG. Notably, the expression of this miRNA was remarkably decreased in patients with higher disease activity. miR-302d has been identified as an important modulator of type I interferon (IFN)-associated gene expression in SLE [44]. Differential expression of seven miRNAs was identified in plasma samples from SLE patients versus healthy controls, including over-expression of miR-142-3p and miR-181a and the downregulation of miR-106a, miR-17, miR-20a, miR-203 and miR-92a [45]. Moreover, significant downregulation of miR-342-3p, miR-223, and miR-20a was also reported in patients with active nephritis. The peripheral expression of miRNAs might be regarded as a diagnostic biomarker for SLE. The miRNAs mentioned were predicted to target TGF-β-related elements in addition to those regulating apoptosis, cytokine–cytokine interactions and T cell differentiation [45]. Downregulation of plasma miR-200b-5p, miR-141-5p, and miR-200c-5p was found in patients with lupus nephritis compared with healthy individuals [46]. The diagnostic values of these miRNAs were estimated to be 0.748, 0.74 and 0.723, respectively. The combination of these three miRNAs increased the diagnostic value of lupus nephritis to 0.936. Finally, the plasma level of miR-141-5p was found to be inversely correlated with serum creatinine and the Systemic Lupus Erythematosus Disease Activity Index (SLEDAI ) score, while miR-200c-5p and miR-200b-3p levels were inversely correlated with the SLEDAI score and proteinuria, respectively [46].

Figure 2 shows the molecular mechanism of the participation of a number of miRNAs in the pathogenesis of SLE.

Table 4 and Table 5 summarize the results of studies that have reported the down- and upregulation of miRNAs in SLE samples, respectively.

## 4. Discussion

The highly heterogeneous nature of SLE in terms of both the underlying pathogenic processes and the manifestation of the disease course has hampered a comprehensive understanding of SLE etiology [13]. Recent observations regarding the dysregulation of non-coding RNAs in SLE patients raise an opportunity to clarify the pathobiology of this disorder. The advent of high throughput sequencing technologies and the availability of these data in public databases have facilitated the recognition of the complicated interaction network between lncRNAs, miRNAs and protein-coding genes. A representative study in this regard has identified numerous pairs of lncRNAs and target mRNAs [21]. Such studies have enabled the recognition of therapeutic targets for SLE. Non-coding RNAs have been shown to influence several crucial pathways in the pathogenesis of SLE, including B cell activation and the NF-κB, IFN and TGFβ signaling pathways. STAT-related pathways have been among the most frequently dysregulated pathways in SLE patients. These patients have high IL-17 levels and Th17 numbers. Notably, STAT3 signaling has a critical role in Th17 differentiation [71]. Several miRNAs listed above regulate the STAT3 pathway. Moreover, there is a complex interaction between lncRNAs and miRNAs in the context of SLE, which should be considered in the design of targeted therapies. Several online tools such as StatBase v2.0 (https://web.archive.org/web/20110222111721/http://starbase.sysu.edu.cn/) and TargetScan v7.2 (http://www.targetscan.org/vert_72/) can be used for the prediction of miRNA targets. Non-coding RNA signatures, particularly miRNAs, are not only useful for the discrimination of SLE patients from healthy subjects, but could also potentially distinguish the presence of SLE-related complications [20].

Although the association between SNPs within coding regions and the risk of SLE has been assessed in several studies [72], the contribution of genomic variants within non-coding regions has been less well explored. Based on the crucial roles of non-coding RNAs in the pathogenesis of SLE and the prominent role of SNPs in the regulation of their functions, these SNPs are regarded as putative risk variants for SLE. Thus, better identification of the role of these variants might lead to the identification of the underlying cause of variability in the disease course or the response of patients to therapeutic modalities.

Overall, despite extensive research in the field of non-coding RNAs’ participation in SLE, this research avenue has not reached the level of clinical application except for a small number of studies that have verified the diagnostic/prognostic roles of these transcripts in SLE. Animal studies have raised the hope that modification of the expression of these transcripts could influence the disease course. Thus, future studies in this field may facilitate the identification of new treatment modalities for SLE.

## Figures and Tables

**Figure 1 biomolecules-10-00937-f001:**
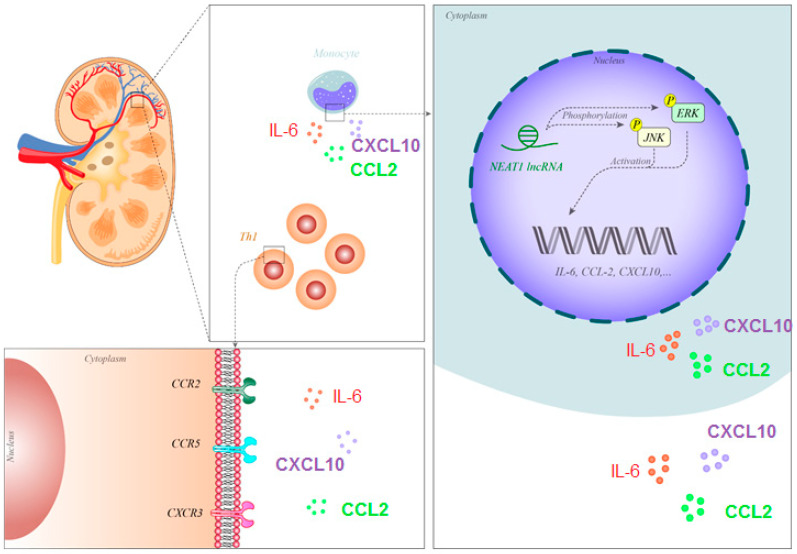
NEAT1 has been shown to be over-expressed in the monocytes of SLE patients. This lncRNA phosphorylates JNK and ERK, thus activating these proteins and enhancing the expression of IL-6, CCL2 and CXCL10. These cytokines/chemokines attract Th1 cells, thus participating in the pathogenesis of nephritis [22]. Abbreviations: SLE, systemic lupus erythematosus; lncRNA, long non-coding RNA, Il-6: interleukin-6, CCL2: C-C motif ligand 2, CXCL10: C-X-C motif chemokine 10.

**Figure 2 biomolecules-10-00937-f002:**
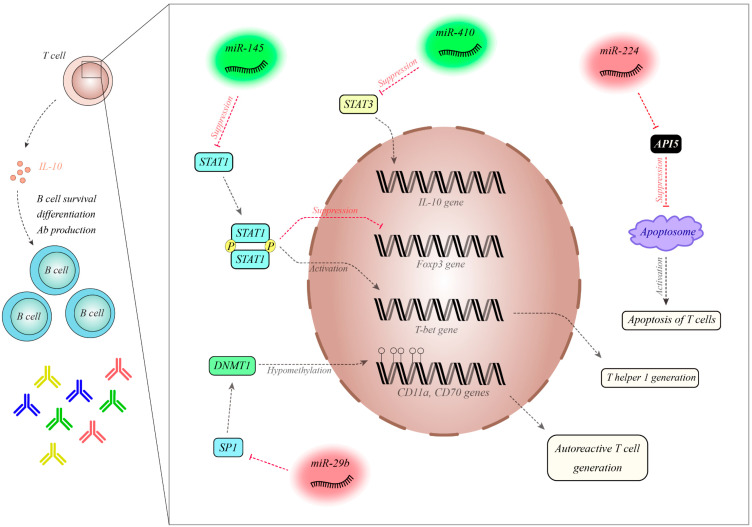
MiR-145 is decreased in SLE patients. MiR-145 has a role in the inhibition of STAT1, which is involved in the differentiation of Treg cells via its inhibition of Foxp3. Moreover, STAT1 induces T-bet, which participates in the differentiation of Th1 cells [36]. MiR-410 is downregulated in SLE patients and is a suppressor of STAT3. STAT3 is an inducer of IL-10. The levels of this cytokine are increased in SLE patients, leading to the enhanced survival and differentiation of B cells and the increased production of antibodies [37]. MiR-224 is increased in SLE patients and has been shown to inhibit API5. API5 decreases the expression of APAF1 and precludes the activation of caspases, thereby hampering apoptosome formation. Thus, the over-expression of miR-224 leads to increased T cell activation-induced cell death [36]. MiR-29a is upregulated in SLE patients and suppresses Sp1, which is an inducer of ANMT1. The over-expression of miR-29a leads to hypomethylation of CD11a and CD70, which are involved in the production of autoreactive T cells [38] Abbreviations: Treg cell: Regulatory T cell, DNMT1: DNA (cytosine-5)-methyltransferase 1, SP1: specificity protein 1.

**Table 1 biomolecules-10-00937-t001:** List of lncRNAs whose expression has been downregulated in SLE patients (PBMC: peripheral blood mononuclear cell, SPF BALB/c mice: specific pathogen-free (SPF) bcl-2 transgenic BALB/c mice).

LncRNA	Number of Clinical Samples (Tissues, Serum, etc.)	Assessed Cell Line	Targets/Regulators	Signaling Pathways	Function and Comments	Reference
lnc-HSFY2-3:3	15 SLE patients and 15 healthy donors were enrolled.	Monocyte-derived dendritic cells (moDCs)				[18]
lnc-SERPINB9-1:2						
GAS5	24 SLE patients and 12 healthy controls were enrolled.	Plasma			GAS5 induces apoptosis and growth arrest in human peripheral blood T cells and increases apoptosis.	[20]
lnc-DC			STAT3		Lnc-DC, which is expressed exclusively in dendritic cells, supports their capacity to stimulate T cell activation.	
PRSS30P	5 SLE patients and 6 age- and gender-matched normal individuals were enrolled in this study.	Plasma				[21]
RP11-23P13.6						
ARRDC1-AS1						
RP4-758J18.2			CCNL2		The downregulation of RP4-758J18.2 and CCNL2 may play a key role in the immune response of SLE.	
RP11-66N24.3						
LINC00996			REPIN1			
RP11-649A18.12			NUP85, MIF4GD			
RP11-539L10.2			MAN2B2			
RP11-23J9.5						
TMEM147-AS1						
lnc5150	76 SLE patients and 71 healthy controls were recruited in this study.	PBMCs			Lnc5150 might disclose a diagnostic value for SLE.	[23]
linc0949	Samples were obtained from 102 SLE patients and 76 healthy donors.	PBMCs	TNF-α, IL-6		Linc0949 could be a potential biomarker for diagnosis, disease activity and therapeutic response in SLE.	[24]
CTC-471J1.2	Samples were collected from 29 SLE patients and 34 age- and sex-matched healthy control subjects.	PBMCs				[19]
TUG1	Samples from 85 SLE patients and 95 healthy controls were collected.	PBMCs	miR-223		TUG1 might be associated with inflammation and renal injury in SLE pathogenesis.	[26]
	A total of 24 SPF BALB/c female mice (8 weeks old) (SLE models: *n* = 18, controls: *n* = 6).	Blood samples, kidney tissue	p65	NF-κB signaling pathway	Inhibition of NF-κB signaling might be a potential target in SLE treatment.	[25]
SLEAR	130 SLE patients and 119 healthy controls were enrolled.	Jurkat cells	ILF2, hnRNP F and TAF15		SLEAR expression negatively correlates with degree of apoptosis in SLE patients.	[28]
uc001ykl.1	24 SLE patients and 12 age-matched healthy volunteers were recruited.	T cells	CDK6		A decreased uc001ykl.1 level may induce CRISPLD2 downregulation and then result in inflammation and abnormalities of the erythrocyte sedimentation rate (ESR) and CRP.	[27]
ENST00000448942			TNFSF10, TNFAIP3		The correlation between ENST00000448942 and ESR may be due to its potential role in positively regulating expression of TNFAIP3, an important regulator of inflammation.	

**Table 2 biomolecules-10-00937-t002:** List of lncRNAs whose expression has been upregulated in SLE patients.

LncRNA	Number of Clinical Samples (Tissues, Serum, etc.)	Assessed Cell Line	Targets/Regulators	Signaling Pathways	Function and Comments	Reference
TSIX	15 SLE patients and 15 healthy donors were enrolled.	Monocyte-derived dendritic cells (moDCs)	collagen mRNA		TSIX is a new regulator of collagen expression which stabilizes the collagen mRNA.	[18]
NRIR	5 SLE patients and 6 age- and gender-matched normal individuals were enrolled in this study.	Plasma	RSAD2	IFN pathway	NRIR is a negative regulator of the IFN pathway and other signaling pathways that are involved in innate and adaptive immunity.	[21]
AC002511.3			FFAR3			
CTD-2313J17.1			FAM174B			
RP11-196G18.3			FCGR1A, HIST2H2BE, BOLA1			
RP11-561O23.8						
AC005532.5			C1GALT1			
TMLHE-AS1						
AC007556.3						
RP11-153M7.5			TLR2		Increased expression of RP11-153M7.5 and TLR2 may promote thrombosis and autoantibody production, which contributes to the process of SLE.	
RP11-242C19.2						
linc0597	24 SLE patients and 12 healthy controls were enrolled.	Plasma	TNF-α, IL-6		Linc0597 is involved in innate immunity and regulates the induction of proinflammatory cytokines.	[20]
MALAT1	36 SLE patients and 45 age-matched and sex-matched normal controls were recruited.	PBMCs	IL-21, SIRT1	SIRT1 signaling	MALAT-1 exerts its detrimental effects by regulating SIRT1 signaling in both THP-1 cell lines and human primary monocytes.	[29]
RP11-875O11.1	Samples were collected from 29 SLE patients and 34 age- and sex-matched healthy control subjects.	PBMCs				[19]
NEAT1	Samples from 28 SLE patients and 28 healthy donors were collected.	PBMCs	BAFF	NEAT1-BAFF pathway	NEAT1 may act as an important component of the molecular circuitry to regulate the activation of the immune system.	[30]
	29 patients with SLE and 40 age-matched and sex-matched normal controls were recruited.	PBMCs	IL-6, CXCL10	MAPK pathway	NEAT1 participates in the pathogenesis of SLE through dysregulating proinflammatory chemokines and cytokines by affecting the TLR4-mediated inflammatory pathway.	[22]
RP11-2B6.2	Samples were collected from 22 SLE patients suffering from lupus nephritis and 7 healthy controls.	Renal tissue	SOCS1	IFN-I signaling pathway	Knockdown of lncRNA RP11-2B6.2 inhibits the phosphorylation of JAK1, TYK2, and STAT1 in IFN-I pathway, while promoting chromatin accessibility and the transcription of SOCS1.	[31]
linc00513	22 SLE patients and 7 healthy controls were recruited.	Renal tissue	STAT1, STAT2	IFN signaling pathway	Linc00513 is responsible for amplified IFN signaling in SLE patients.	[32]

**Table 3 biomolecules-10-00937-t003:** Variants within lncRNAs which are associated with risk of SLE.

LncRNA	Number of Clinical Samples	Risk Variant	Reference
SLEAR	4,556 SLE patients and 9,451 healthy controls were enrolled.	rs13259960	[28]
linc00513	139 SLE patients were recruited for genotyping.	rs205764	[32]
		rs547311	

**Table 4 biomolecules-10-00937-t004:** List of miRNAs whose expression has been downregulated in SLE patients.

miRNA	Number of Clinical Samples	Assessed Cell Line	Targets/Regulators	Signaling Pathways	Function and Comments	Reference
miR let-7f	In total 15 SLE patients and 10 normal controls were enrolled.	Bone marrow-derived mesenchymal stem cells (BM-MSCs)/PBMCs	IL-6	STAT3 signaling pathway	MiR let-7f increases the apoptosis rate of BM-MSCs through targeting IL-6 and activating the STAT3 pathway.	[39]
miR-31	MRL/lpr mice and control MRL/MpJ mice were purchased from the Jackson Laboratory (JAX, Maine, USA)	PBMCs	Foxp3, serine/threonine kinase 40, IL-2	NF-κB signaling pathway	MiR-31 enhances inflammatory cytokine production.	[33]
miR-98	Samples from 41 SLE patients and 20 healthy controls were collected.	PBMCs	IL-6	IL-6/STAT3 signaling pathway	MiR-98 could ameliorate STAT3-mediated cell proliferation and inflammatory cytokine production.	[40]
	Samples from 48 SLE patients and 39 healthy controls were collected.	PBMCs	Fas	Fas-mediated apoptosis	MiR-98 downregulation contributes to the dysregulation of apoptosis in SLE.	[41]
miR-125a	Samples were gathered from 10 SLE patients and 8 healthy donors.	PBMCs	KLF13, RANTES	Inflammatory chemokine pathway	MiR-125a acts as a negative regulator in the feedback loop of RANTES expression in activated T cells.	[42]
miR-125b	50 SLE patients and 26 healthy controls were enrolled in this study.	PBMCs	TNF-α, ETS1, STAT3	NF-κB signaling	MiR-125b expression correlates with LN and the dysfunction of T cells by affecting the target genes ETS1 and STAT3.	[43]
miR-302d		PBMCs	IRF9	IFN pathway	Inhibition of the type I IFN pathway through the manipulation of miR-302d levels could be beneficial in SLE patients who present with clinical disease driven by IFN dysregulation.	[44]
miR-106a	68 SLE patients and 68 healthy controls were enrolled.	Plasma	CREB1, TGFBR2	TGFβ signaling pathway	These control monocytopoiesis and regulate regulatory T cells.	[45]
miR-20a						
miR-17			Bim, PTEN, CREB1, TGFBR2		These regulate apoptosis, control monocytopoiesis and regulate regulatory T cells.	
miR-20a						
miR-92a						
miR-203			TGFBR1/2, ACVR2A/2B, SMAD6/7, SMURF1, BMPR2, MAPK1		This miRNA targets genes in the MAPK signaling and cytokine–cytokine receptor pathways and many genes involved in focal adhesion and tight junctions.	
miR-342-3p					The association of miR-223 and miR-342-3p with lupus nephritis was observed in this study.	
miR-223						
miR-200b-5p	101 SLE patients suffering from LN and 100 healthy controls were enrolled.	Plasma	IKBKB, IL-8, IL-6, NF-kB	Type I IFN, PTEN, KLK4 and SOCS1	The miR-200 family could target several genes and pathways to inhibit inflammatory responses and kidney fibrosis, thereby decreasing renal damage.	[46]
miR-200c-5p						
miR-141-5p						
miR-103	50 SLE patients and 30 healthy controls were involved in this study.	Plasma			The signature of circulating miRNAs will provide novel biomarkers for the diagnosis of SLE and the evaluation of disease activity and LN.	[34]
miR-15b						
miR-19b						
miR-22						
miR-23a						
miR-93						
miR-654	Samples from 24 SLE patients and 24 controls were collected.	Serum	ERK, AKT, IL-1β, IL-6, IL-8, TNF-α	MIF-dependent pathways	MiR-654 inhibits MIF (migration inhibitory factor) expression via binding to MIF 3’ UTR and reduces downstream inflammatory cytokine production.	[47]
miR-124	27 SLE patients and 34 healthy controls were recruited in this study.	Serum		TNF, TGF-β, NF-κB and MAPK signaling pathways	This miRNA may have regulatory effects on immunity by affecting signaling pathways, and may represent a specific biomarker for distinguishing patients with autoimmune diseases from healthy controls.	[48]
miR-146a	52 SLE patients and 29 healthy controls were recruited.	Serum	IRAK1, TRAF6, IRF-5, STAT-1	Type I IFN pathway	The under-expression of miR-146a in lupus patients is relevant to the biological and clinical behavior of SLE.	[49]
miR-1246	A total of 50 SLE patients and 20 healthy controls were enrolled.	B cells	EBF1	AKT-P53 signaling pathway	Therapies that turn the expression of affected miR-1246 genes back to normal could serve as a potential and effective method for treating SLE.	[50]
miR-377	42 SLE patients and 48 healthy controls contributed to the study.	T cells		Vitamin D signaling pathway	Severe vitamin D deficiency is associated with decreased observed miRNA levels in SLE patients. VDR mRNA expressions in the T cells of SLE patients were significantly lower than those in controls, but CYP24A1 and CYP27B1 mRNA levels were significantly increased.	[51]
miR-342						
miR-10a						
miR-374b						
miR-410						
miR-410	20 SLE patients and 20 healthy subjects were recruited.	T cells	STAT3	STAT3 signaling pathway	MiR-410 is the key regulatory factor in the pathogenesis of SLE, which regulates the expression of IL-10.	[37]
miR-145	Samples from 26 SLE patients and 27 healthy controls were collected.	T cells	STAT1	Interferon-mediated signaling pathway	The over-expression of miR-145 suppresses the gene expression of STAT-1 which seems to be associated with lupus nephritis.	[36]
miR-23b	18 SLE patients suffering from LN and 9 healthy controls were enrolled.	Kidney biopsy	TAB2, TAB3, IKK-α	NF-κB pathway	The over-expression of miR-23b suppresses both TNF-α- and IL-1β-induced NF-κB activation.	[52]
hsa-miR-371-5p	35 SLE patients suffering from LN and 35 healthy controls were included in this study.	Kidney biopsy	HIF-1α		The over-expression of hsa-miR-371-5p may inhibit mesangial cell proliferation and promote apoptosis.	[53]

**Table 5 biomolecules-10-00937-t005:** List of miRNAs whose expression has been upregulated in SLE patients (LPS: lipopolysaccharide, DC: dendritic cell, NK: natural killer).

miRNA	Number of Clinical Samples	Assessed Cell Line	Targets/Regulators	Signaling Pathways	Function and Comments	Reference
miR-663	A total of 13 SLE patients and 10 healthy controls were enrolled.	BM-MSCs	TGF-β1, JUNB and JUND, MYL9, GRID2D, EPHB3	P38/MAPK and Akt signaling pathways	The inhibition of miR-663 in BMSCs could restore the production of Treg and Tfh cells through the secretion of TGF-β1.	[54]
miR-155	24 SLE patients and 75 healthy donors were enrolled.	Serum	SHIP1, SOCS1, EKKε, TAB2, S1pr1, IL4, IL-17A	U1-RNP/TLRs/IFNs signaling pathway	MiR-155 regulates the distribution of T cells.	[55]
miR-143			TLR2, NFκB		MiR-143 reduces the production of TLR4 and pro-inflammatory cytokines (IL-1β, IL-6, and TNF-α).	
miR-132			IRAK4		Monocytes and macrophages show an increased expression of miR-132 in response to LPS.	
miR-126			TRAF6, IFN		MiR-126 inhibits the release of proinflammatory cytokines.	
miR-29a			IFN-γ		MiR-29a contributes to inhibiting the response to intracellular bacterial infections.	
miR-448	27 SLE patients and 34 healthy controls were recruited in this study.	Serum		TNF, TGF-β, NF-κB and MAPK signaling pathways	These miRNAs may have regulatory effects on immunity by affecting signaling pathways, and may represent specific biomarkers for distinguishing patients with autoimmune diseases from healthy controls.	[48]
miR-551b						
miR-130b-3p	60 SLE patients suffering from LN and 30 healthy controls were recruited.	Serum	ERBB2IP, PGC-1α, PPAR-γ	TGF-β-induced signaling pathways	The circulating miR-130b-3p might serve as a biomarker and play an important role in renal damage in early-stage LN patients.	[56]
miRNA-371b	Samples were collected from 1092 SLE patients and 553 healthy controls.	Serum			It was suggested that the microRNAs mentioned may be associated with aberrant functions of CD4+ and CD8+ T cells in SLE.	[57]
miRNA-5100						
hsa-miR-30e-5p	70 SLE patients and 40 healthy controls were included.	Plasma	Granzyme B, perforin		MiR-30e expression is suppressed by IFN-α, which in turn suppresses natural killer cell cytotoxicity by targeting granzyme B and perforin.	[58]
miR-142-3p	68 SLE patients and 68 healthy controls were enrolled.	Plasma	TGFBR1	TGFβ signaling pathway	Increased miR-142-3p expression may be caused by increased cellular release of this miRNA through increased exocytosis and/or the normal exocytosis of cells containing increased miRNA.	[45]
miR-181a					This miRNA is important for hematopoietic cell differentiation, increases the fraction of B-lineage cells, and is also expressed by endothelial cells.	
miR-132	Human THP-1, HEK293, and murine RAW264.7 cells were obtained from the American Type Culture Collection.	PBMCs	IRAK4	TLR-signaling pathway	MiR-132 was shown to regulate neuronal morphogenesis and the dendritic plasticity of cultured neurons.	[59]
miR-212					MiR-212 can interfere with the craving for cocaine in mice and acts as a tumor suppressor.	
miR-25	28 SLE patients and 28 healthy controls were enrolled in this study.	PBMCs	AMPD2	Purine biosynthesis and metabolism	This study suggested that mentioned miRNAs represent novel diagnostic biomarkers, disease activity markers and potential therapeutic targets for SLE.	[60]
miR-1273h-5p				Genes encoding peroxidase, glycosyltransferase, ATP synthase and hydrolase, carbohydrate phosphatase and exoribonuclease		
miR-148a	8 SLE patients and 7 healthy controls were enrolled in this study.	PBMCs	Gadd45α, PTEN, Bim, DNMT, Bcl2l11	DNA methylation pathway	Members of the miR-148 family, including miR-148a, miR-148b and miR-152, are negative regulators of the innate response and Ag-presenting capacity of DCs.	[33,61]
miR-152						
miR-224			STAT-1, API5		Enhanced expression of miR-224 accelerates T cell activation-induced cell death.	
miR-326						
hsa-miR-345	99 SLE patients and 65 healthy controls were enrolled.	PBMCs	IRF8		IRF8 is a transcription factor which has a key role in regulating the differentiation of B cells and promoting their differentiation. It is revealed that IRF8 is inhibited by hsa-miR-345.	[62]
miR-16	30 SLE patients and 20 healthy controls were enrolled in this study.	PBMCs		MAPK pathway	It was proposed that the over-expression of miR-16 in the blood may be associated with the exposure of immune and inflammatory cells to the circulating blood.	[63]
miR-451					Higher circulating levels of miR-451 might be associated with abnormal erythropoiesis.	
miR-21	MRL/lpr mice and control MRL/MpJ mice were purchased from the Jackson Laboratory (JAX, Maine, USA)	PBMCs	DNMT1		MiR-21 is highly expressed in CD41 T cells in patients with SLE and may contribute to DNA hypomethylation in lupus CD41 T cells.	[33,64]
miR-127						
miR-182						
miR-27a	Samples from 37 SLE patients and 17 healthy controls were collected.	PBMC/NK cells	NKG2D		This study highlighted the expression profile of miR-27a and its potential target NKG2D which is of prime importance for NK cell activation in SLE.	[65]
miR-30a		B cells	Lyn	Pathways of B cell activation	A high level of miR-30a might be responsible for the development of B cell hyperactivity, suggesting that it could be involved in the pathogenesis of SLE.	[66]
miR-152-3p	30 SLE patients and 30 healthy controls were recruited.	B cells	BAFF, KLF5		Knockdown of miR-152-3p expression inhibits the self-reactivity of SLE B cells, thereby reducing their autoantibody production.	[67]
miR-29b	36 SLE patients and 28 healthy donors were recruited in this study.	T cells	sp1, DNMT1		MiR-29b reduces DNMT1 expression levels, thereby resulting in DNA hypomethylation and the over-expression of methylation-sensitive genes, and mediates the pathogenesis of SLE.	[38]
miR-451a	Wild-type (WT, *n* = 10), Faslpr/lpr (*n* = 10), and miR-451a−/− Faslpr/lpr (*n* = 10) female C57B/L6 mice were recruited.	Spleen and thymus tissues	IRF8	IFN pathway	The knockout of miR-451a affects the enlargement of the spleen and reduces urine protein content and immune complex deposits.	[68]
miR-150	Formalin-fixed paraffin-embedded kidney specimens from 14 SLE patients suffering from LN and from 2 normal controls were gathered.	Kidney biopsy	SOCS1	Janus kinase/signal transducers	The over-expression or TGF-β1-induced increase of miR-150 directly decreases SOCS1, leading to increases in profibrotic protein production.	[69]
let-7 family	Samples were collected from 98 SLE patients and 47 healthy controls.	Kidney biopsy	TNFAIP3	Let-7-TNFAIP3-NF-κB pathway	The over-expression of let-7 miRNAs leads to the increased phosphorylation and sustained degradation of IκBα and the enhanced phosphorylation of p65.	[70]

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
