# Peer review of "Exploring the Role of Non-Coding RNAs in the Pathophysiology of Systemic Lupus Erythematosus"

_biomolecules, 2020, doi:10.3390/biom10060937_

Round 1

Reviewer 1 Report

This review paper concerns a topic of interest for rheumatologists and immunologists, i.e. the role of non-coding RNAs in the pathophysiology of SLE. The manuscript is well written and organized and covers the present knowledge in the field indicating the role of non-coding RNAs in the pathophysiology and clinical presentation of SLE and suggests their potential role as biomarkers and potential therapeutic targets.

Minor comments: The cytokines/chemockines indicated as colored dots in Fig.1 should be specified.

Author Response

Minor comments: The cytokines/chemockines indicated as colored dots in Fig.1 should be specified.

Response: We specified the cytokines/ chemokines.

Reviewer 2 Report

In this review, Mohammad Taheri et al., discuss the putative implication of miRNA and LncRNA deregulation in the ethiology of lupus. Although this review is of interest and in general well-written, different points should be improved in order to get a complete view of the role that these RNAs could have in this autoimmune disorder.

In general, the authors should be more accurate in their statements. For instance, “These transcripts comprise a large proportion of the whole transcriptome and exert regulatory roles on specific 31 targets or large regions of the genome (8). “ Could the authors indicate What is the exact proportion of miRNA and LncRNA inside the whole transcriptome?

“The interaction between lncRNAs and transcription factors and the subsequent construction of the 40 preinitiation complex can activate or suppress gene transcription (12).” Do the authors mean mRNA translation (PIC is a structure used for the translation process, not the transcription, Am I wrong ?)? Could you correct or better describe what is the PIC described in this review.

It could help to have an additional figure representing the different mechanisms used by Lcn and miRNAs to modulate transcription and translation.

in figure 2, the apoptosome has to be better defined. Please, could the authors describe the molecular target inside the apoptosome?

STAT3 is also involved in Th17 differentiation, could the authors discuss this point for miR-140 because some recent papers showed an important role played by Th17 cells in lupus progression?

Finally, the study of miRNA and LncRNA is very interesting and the authors should recommend some websites to study the molecular targets of miRNAs.

Author Response

In this review, Mohammad Taheri et al., discuss the putative implication of miRNA and LncRNA deregulation in the ethiology of lupus. Although this review is of interest and in general well-written, different points should be improved in order to get a complete view of the role that these RNAs could have in this autoimmune disorder.

In general, the authors should be more accurate in their statements. For instance, “These transcripts comprise a large proportion of the whole transcriptome and exert regulatory roles on specific 31 targets or large regions of the genome (8). “ Could the authors indicate What is the exact proportion of miRNA and LncRNA inside the whole transcriptome?

Response: We added the mentioned point.

“The interaction between lncRNAs and transcription factors and the subsequent construction of the 40 preinitiation complex can activate or suppress gene transcription (12).” Do the authors mean mRNA translation (PIC is a structure used for the translation process, not the transcription, Am I wrong ?)? Could you correct or better describe what is the PIC described in this review.

Response: We explained the mentioned point.

It could help to have an additional figure representing the different mechanisms used by Lcn and miRNAs to modulate transcription and translation.

Response: Unfortunately, it is difficult for us to prepare another Fig.

in figure 2, the apoptosome has to be better defined. Please, could the authors describe the molecular target inside the apoptosome?

Response: We added this point in the figure legend.

STAT3 is also involved in Th17 differentiation, could the authors discuss this point for miR-140 because some recent papers showed an important role played by Th17 cells in lupus progression?

Response: We discussed this point.

Finally, the study of miRNA and LncRNA is very interesting and the authors should recommend some websites to study the molecular targets of miRNAs.

Response: We recommended such websites.